# An Indoor Smart Parking Algorithm Based on Fingerprinting

Silvia Stranieri 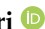

Matematics and Applications, University of Naples Federico II, 80138 Napoli, Italy; silvia.stranieri@unina.it; Tel.: +39-3343980957

**Abstract:** In the last few years, researchers from many research fields are investigating the problem affecting all the drivers in big and populated cities: the parking problem. In outdoor environments, the problem can be solved by relying on vehicular ad hoc networks, which guarantee communication among vehicles populating the network. When it comes to indoor settings, the problem gets harder, since drivers cannot count on classic GPS localization. In this work, a smart parking solution for a specific indoor setting is provided, exploiting the fingerprint approach for indoor localization. The considered scenario is a multi-level car park inside an airport building. The algorithm provides a vehicle allocation inside the car park in quadratic time over the number of parking slots, by also considering the driver's preferences on the terminal to be reached.

**Keywords:** indoor localization; smart parking; fingerprinting

## 1. Introduction

Indoor localization [1] is the process of detecting positions in indoor environments. The big challenge is about not to rely on a global positioning system (GPS), due to the interruption of connection with satellites. In outdoor settings, typically handled through vehicular ad hoc networks [2], localization is an issue from several points of view: from routing [3] to data dissemination [4], but also smart parking [5]. See [6–8] for interesting proposals on localization.

In this work, it has been decided to focus on the last challenging aspect: the smart parking problem. When speaking about the *parking problem*, we mean the problem of handling, in an efficient manner, the process in which drivers compete for a shared resource, namely the available parking slots.

While in outdoor environments the smart parking problem has been widely investigated, the management of such an issue in indoor contexts seems to be still an open issue. For this reason, in this work, the smart parking problem is formalized in an indoor scenario and solved through fingerprinting. The considered scenario is an airport with a car park to be handled. Typically, people approaching the car park simply try to get as close as possible to their target, which in the case of an airport is a terminal, most likely. To the best of my knowledge, there is not any system able to guide us inside a car park in a way that is similar to how GPS guides us along the streets of our cities.

In this work, a fingerprinting localization technique is used. This has been proven to be simple and reliable and it works in two phases: a training offline phase, and an online localization phase [9]. Precisely, fingerprinting is used in this work to measure the received signal strength from any access point, placed in the offline phase, to infer the position of some points of interest, that in the specific airport scenario are the parking slots and the terminals.

Notice that the indoor smart parking problem is nothing but a particular case of indoor localization since it requires finding the position of available slots in an indoor environment. For this reason, the problem presents the same challenges and difficulties as the standard indoor localization problem. It has been decided to focus on the indoor smart

parking problem instead of simple indoor localization since the latter has been already investigated deeply.

*Outline*

The rest of the paper is organized as follows: in Section 2, the state of the art on parking algorithms is analyzed, both for indoor and outdoor settings; in Section 3, some background concepts are introduced, such as RSSI and fingerprinting; in Section 4, the considered environment is explained and in Section 5, some formal definitions and notations are provided; Section 6 shows the indoor parking algorithm and its complexity in all of its phases, by also applying each step to a running example; in Section 4, the algorithm performance is evaluated over several executions with different parameters; finally, in Section 8, the conclusions of this work are provided.

## 2. Related Works

The parking problem is still an open issue for many researchers from diverse research fields. In outdoor contexts, several proposals have been made to provide algorithmic solutions to the problem, such as authors of [10], which produced an algorithm to complete the parking process by optimizing the waiting time and considering the destination of the user when assigning a parking slot to him. Additionally, the authors in [11] provided an algorithm based on a known optimization problem, which is the ant colony optimization, by getting inspired by the behavior of real ants looking for food. The authors of [12], instead, proposed a game-theoretic solution, by modeling the parking problem as a multi-agent game in which drivers compete towards the same resource. In this case, the solution is provided by means of a social equilibrium, the Nash equilibrium.

When moving to an indoor setting, one can observe that the literature on smart parking is much poorer. Nevertheless, some interesting solutions to the problem has been proposed, such as the authors in [13] which provided a vision-based parking lot system, where wide-angle fisheye-lens are used and in which vacant parking spaces can be detected by a background subtraction scheme. However, in [14], the authors provided a system based on smartphones that tracks the vehicle's location in real time and uses landmark recognition methods against noises. Additionally, in [15], they proposed a new map representation for indoor localization: precisely, they considered the permanent elements as static objects and the changing elements as semi-static objects. Recently, the authors of [16] developed a car-searching mobile app based on a turn-by-turn navigation strategy able to correct the user's heading orientation.

In [17], the authors also used fingerprinting applied to smart parking, with the aim of recording parking positions, through the K-nearest neighbor and fingerprinting techniques based on the RSSI Bluetooth with distance calculation parameters using Manhattan distance. However, they did not formalize and face the problem of finding a suitable parking slot in an indoor context. To the best of my knowledge, this is the first work addressing the indoor parking problem, by providing a solution to associate vehicles to available parking slots by getting drivers as close as possible to their final destination, by means of a fingerprinting-based localization technique.

## 3. Background

*3.1. Outdoor and Indoor Localization*

Nowadays, in outdoor environments, the global position system constitutes an established instrument to localize vehicles, pedestrians, and so on. When one moves to indoor contexts, one has to deal with several challenging aspects, as they say in [18]. Indeed, the continuous interruption of connection with satellites makes indoor localization much harder. See [19] for an overview of the existing localization system technologies and algorithms. The classical positioning techniques used for indoor environments are [18]:

- Time of arrival: based on the computation of the time needed by the signal to go from the unlocated device (UD) to the base station. To detect the position of UD, three base stations are needed.
- Time difference of arrival: it evaluates the time difference at which the signal arrives at many measuring units.
- RSS-based fingerprinting: after a collection phase of several RSS fingerprints, it is based on matching online measurements with the closest possible location in the database.
- Angle of arrival: based on the computation of the angle at which the signal arrives from the UD to the base station.

*3.2. RSSI*

The received signal strength indicator (RSSI) measures the power received in a radio signal [20]: the greater the RSSI the stronger the signal. The RSSI values are measured in dBm and have typical negative values ranging between 0 dBm (excellent signal) and −110 dBm (extremely poor signal) [21]. We call RSS values the measurements associated with radio signals.

*3.3. Fingerprinting*

In general, in the same way, a fingerprint identifies uniquely humans, in computer science fingerprints are used to map a large data representation to a much shorter one. In the specific scenario of the localization, given a map, the set of RSS values that are collected for each position in the map from various base stations is called the *fingerprint* for that location. Matching the observations of RSS to the map of the previously measured RSS values is known as *fingerprinting* [22]. See [23,24] for interesting proposals on fingerprinting.

**4. Environment Setting**

For this work, let us imagine a scenario as the following: a multi-level car park within an airport. Clearly, as for the outdoor case, vehicles looking for a free space should be communicating with each other so as to ensure a reduction of traffic congestion and air pollution, but also to facilitate drivers in finding a parking slot which is suitable for his requirements.

While in an outdoor setting drivers can exploit vehicular ad hoc networks to communicate with each other, in an indoor environment things get harder. Indeed, the idea is to perform an indoor localization of vehicles by relying on a slightly different version of fingerprinting approach.

As you can see from Figure 1, the car park building presents several access points, used to measure the signal strength received from the red points, whose position has to be discovered. Those points should be placed in the proximity of each parking slot, so as to know their position in the building.

Some interesting points should be also considered as possible destination points for drivers. More precisely, in a typical scenario, drivers looking for a free parking slot want to allocate their car and then reach some destination. In our airport setting, the interesting destination could easily be the terminals for departure. They are denoted by T in Figure 1.

Moreover, the car park is supposed to be divided into areas with the final aim of associating drivers with an available parking slot belonging to the same area of the target terminal.

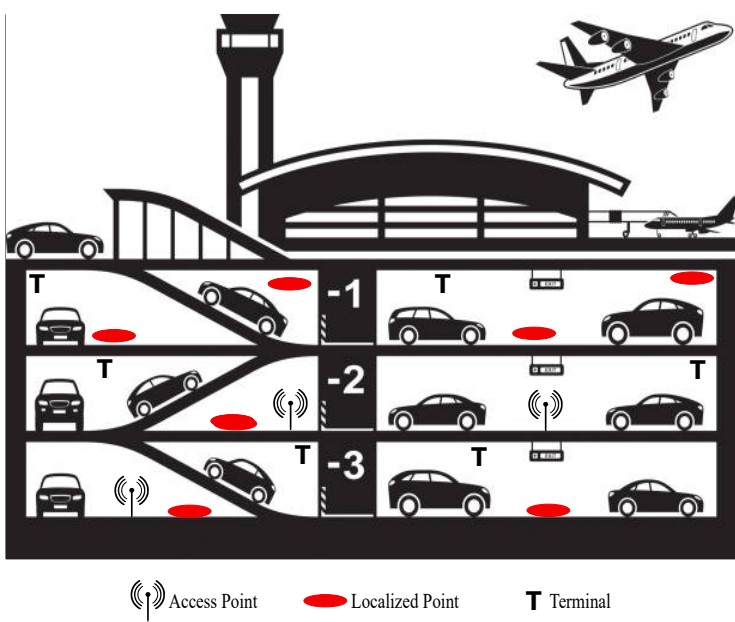

**Figure 1.** Example of environment setting: a multi-level car park at the airport.

## 5. Notation

In this section, some formal definitions are given to define the indoor parking problem. First, let us fix some notation:

- $AP$ is the set of access points, with $|AP| = n$;
- $S$ is the set of slots in the considered environment, with $|S| = m$;
- $A$ is the set of areas of the car park, with $|A| = k$;
- $T$ is the set of terminals of the airport, with $|T| = l$;
- $rs_i, \forall i \in AP$ is the received signal strength associated with the $i$-th access point in $AP$;
- $rs_j, \forall j \in T$ is the received signal strength associated with the $i$-th terminal in $T$;
- $av : S \to \{0, 1\}$ is the *available* function associating to each slot a 0 value if it is taken, or a 1 value if it is available for parking;
- $area : AP \to A$ is the function associating the belonging area in the car park to each access point;
- $is : S \cup T \to A$ is the function associating the belonging area in the car park to each slot or terminal, which will be defined in the sequel.

Let us move to defining the actual fingerprint definition.

**Definition 1** (Fingerprint). *Given a slot $s \in S$, let us denote by $fp_s = \langle rs_1, \ldots, rs_n \rangle$ the associated fingerprinting tuple. Equivalently, given a terminal $t \in T$, let us denote by $fp_t = \langle rs_1, \ldots, rs_l \rangle$ the associated fingerprinting tuple.*

Notice that some issues related to the unreliability of the RSS values could be easily faced by introducing a mechanism similar to the one of DGPS (differential GPS) [25]. Indeed, one can use one or more access points as detectors of variations to be transmitted through the network to improve the quality of the measurements, such as they did in [26] and in [27].

We denote by $max(fp_i)$ the maximum entry of the $fp_i$ tuple, indicating the $ap \in AP$ with the highest received signal strength for the slot (terminal) $i$ (namely, the closest access point with respect to $i$). Precisely, $is(i) = area(max(fp_i))$.

To better define the notion of *area*, let us introduce a parameter, $\theta$, representing the maximum allowed difference between the signal strengths of the fingerprint entries related to the closest access point.

**Definition 2** (Parking Area). *Given $fp_i$ the fingerprint tuple associated to the parking slot i defined as in Definition 1, and given a threshold θ, a parking area is made of those parking slots i and j such that:*

$$max(fp_i) - max(fp_j) \leq \theta \ \wedge \tag{1}$$

$$max(fp_i) \text{ and } max(fp_j) \text{ occupy the same position in the corresponging fingerprints} \tag{2}$$

*The same holds in the case of terminals in place of slots.*

## 6. Indoor Parking Algorithm

In this section, the phases of the indoor parking algorithm are described through a running example.

### 6.1. Sampling Phase

In this offline phase, the airport car park is sampled. Precisely, the access points are placed in the environment in a way that the whole car park is covered by the signal. Let us consider the example shown in Figure 2: the scenario presents four terminals, three access points and twelve parking slots among the several parking areas, among which the black ones are already taken. In this example, the sampling phase produced the access points distribution shown in the figure.

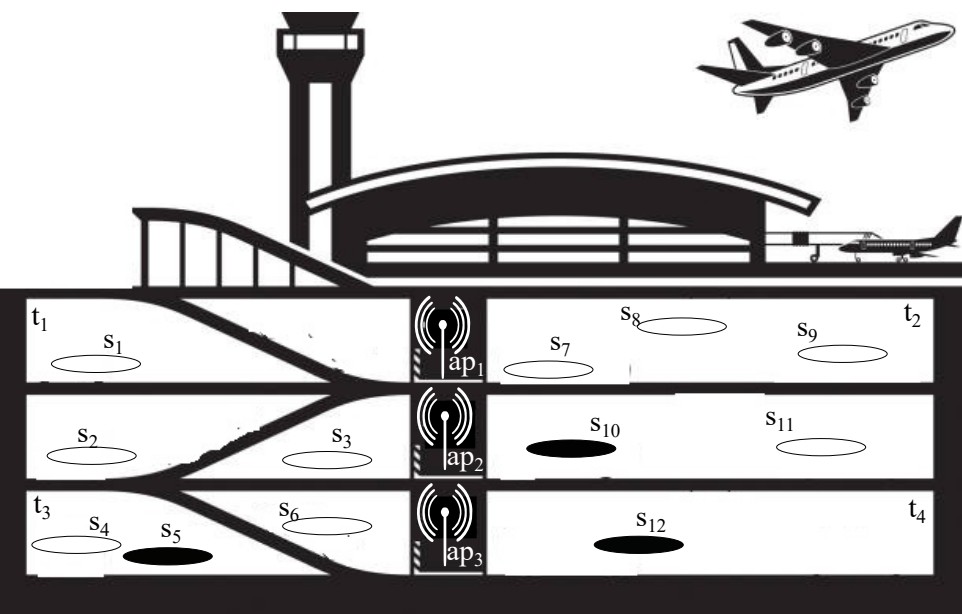

**Figure 2.** A running example.

### 6.2. User Profiling

Since we are dealing with drivers looking for parking space in an airport building, without loss of generality, it is possible to assume that those drivers have to take a plane, or they have to pick someone up who is arriving by plane.

For this reason, one can assume a user profiling phase, in which drivers provide a flight number. Such information allows the system to infer the terminal number that the driver wants to reach.

This will be useful to execute the algorithm according to the driver's needs, so to associate each driver with the parking slot which best suits his requirements in terms of the destination to be reached.

### 6.3. Fingerprint Construction

In this phase, the fingerprint of each slot and terminal is built according to Definition 1, by following Algorithm 1.

---
**Algorithm 1** Fingerprint Construction

---
**Input:** $AP, T, S$
**Output:** $fp$
  **for** $s \in S$ **do**
    **if** $av(s)$ **then**
      **for** $ap \in AP$ **do**
        $fp_s[ap] = receivedSS(s, ap)$
      **end for**
    **end if**
  **end for**
  **for** $t \in T$ **do**
    **for** $ap \in AP$ **do**
      $fp_t[ap] = receivedSS(t, ap)$
    **end for**
  **end for**

---

The sets $AP, T, S$ are the once defined in Section 5 and the function *receivedSS* extracts the signal strength from the access point to the desired interesting point.

By considering the running example in Figure 2, the resulting fingerprints are shown in Tables 1 and 2.

**Table 1.** Slot fingerprints.

| | |
|---|---|
| $fp_{s_1}$ | $\langle 0.9, 0.6, 0.3 \rangle$ |
| $fp_{s_2}$ | $\langle 0.5, 0.8, 0.2 \rangle$ |
| $fp_{s_3}$ | $\langle 0.6, 0.9, 0.3 \rangle$ |
| $fp_{s_4}$ | $\langle 0.1, 0.5, 0.7 \rangle$ |
| $fp_{s_6}$ | $\langle 0.2, 0.6, 0.9 \rangle$ |
| $fp_{s_7}$ | $\langle 1, 0.7, 0.2 \rangle$ |
| $fp_{s_8}$ | $\langle 0.9, 0.6, 0.1 \rangle$ |
| $fp_{s_9}$ | $\langle 0.8, 0.5, 0 \rangle$ |
| $fp_{s_{11}}$ | $\langle 0.4, 0.6, 0.4 \rangle$ |

**Table 2.** Terminal fingerprints.

| | |
|---|---|
| $fp_{t_1}$ | $\langle 0.5, 0.3, 0.1 \rangle$ |
| $fp_{t_2}$ | $\langle 0.5, 0.3, 0.1 \rangle$ |
| $fp_{t_3}$ | $\langle 0.1, 0.3, 0.5 \rangle$ |
| $fp_{t_4}$ | $\langle 0.1, 0.3, 0.5 \rangle$ |

### 6.4. Area Construction

In this phase, the entire environment is divided into areas in such a way that slots and terminals belonging to the same area have the same closest access point.

While assigning an area to each slot and terminal, it could be necessary to update the value of the threshold $\theta$ so as to have that each terminal has at least one parking slot in the same area.

Based on the parking area definition in Definition 2, by using a threshold of 0.2, one would obtain the areas of Table 3. The result is clearly unsatisfactory: indeed, all the terminals do not fall in any area containing some parking slots. For this reason, the areas are computed again by increasing the value of the threshold, for instance, 0.4. The results are shown in Table 4.

**Table 3.** Area construction with $\theta = 0.2$.

| | |
|---|---|
| $a_1$ | $s_1, s_7, s_8, s_9$ |
| $a_2$ | $s_2, s_3, s_{11}$ |
| $a_3$ | $s_4, s_6$ |
| $a_4$ | $t_1, t_2$ |
| $a_5$ | $t_3, t_4$ |

**Table 4.** Area construction with $\theta = 0.4$.

| | |
|---|---|
| $a_1$ | $s_1, s_7, s_8, s_9, t_1, t_2$ |
| $a_2$ | $s_2, s_3, s_{11}$ |
| $a_3$ | $s_4, s_6, t_3, t_4$ |

The algorithm performing the computation explained above can be summarized in Algorithm 2.

---

**Algorithm 2** Area Construction

---

**Input:** $\theta, S, T$
**Output:** $A$
  **for** $s \in S$ **do**
    $associate\_area(s, \theta)$
  **end for**
  **for** $t \in T$ **do**
    $associate\_area(t, \theta)$
  **end for**
  **while** $\exists t \in T$ s. t. $!contains(area(t), s)$ for some $s \in S$ **do**
    update $\theta$ and start over
  **end while**

---

The function *associate_area* is supposed to associate a slot or a terminal to an existing or a new parking area according to the Definition 2, while the function *contains*$(a, b)$ returns true if $b$ is contained in $a$.

### 6.5. The Driver-Slot Association

In order to complete the indoor parking process, we also need to take into consideration the set of drivers $D$ ready to park. As previously said, each driver can express the terminal he wants to reach after leaving his car during the user profiling phase.

By following the running example, let us consider $D = \{d_1, d_2, d_3, d_4, d_5, d_6\}$ the set of drivers and a function *terminal* $: D \rightarrow T$ associating to each driver his favorite terminal.

Table 5 provides an example with 6 drivers and their preferences.

**Table 5.** Example of driver's preferences.

| | **Favourite Terminal** |
|---|---|
| $d_1$ | $t_1$ |
| $d_2$ | $t_1$ |
| $d_3$ | $t_3$ |
| $d_4$ | $t_4$ |
| $d_5$ | $t_2$ |
| $d_6$ | $t_2$ |

Algorithm 3 produces a set *Result* $\subseteq D \times S$ made of pairs $(driver, slot)$ such that the slot has been associated with the driver, according to his preferences.

---

**Algorithm 3** Slot Association

---

**Input:** $D, A$
**Output:** *Result*
  **for** $d \in D$ **do**
    $area = is(terminal(d))$
    $s = get\_available\_slot(area)$
    **if** $!s$ **then**
      $area = get\_next\_available\_area(A)$
      **if** $!area$ **then**
        *break*
      **end if**
      $s = get\_available\_slot(area)$
    **end if**
    **if** $!s$ **then**
      *exit*
    **end if**
    $add(Result, (d, s))$
    $remove(s, area)$
  **end for**

---

The function $get\_available\_slot(area)$ gives a random available slot in the selected area. The function $get\_next\_available\_area(A)$ simply returns the next area in $A$ having at least one available parking slot (if any). The function *add* allows putting the new pair $(driver, slot)$ in the resulting set. The function $remove(s, area)$ simply makes the slot $s$ no longer available.

The resulting set by applying Algorithm 3 to the running example is shown in Table 6. They are obtained by applying the algorithm to the areas obtained in Table 4.

**Table 6.** Resulting set of the running example.

| Driver | Slot |
|:------:|:----:|
| $d_1$ | $s_1$ |
| $d_2$ | $s_7$ |
| $d_3$ | $s_4$ |
| $d_4$ | $s_6$ |
| $d_5$ | $s_8$ |
| $d_6$ | $s_9$ |

Since after this execution, $a_1$ and $a_3$ run out of available parking slots, what would happen if some other driver showed up? Let us suppose of having two additional drivers $d_7$ and $d_8$ whose preferences about the target terminal are $t_3$ and $t_4$ respectively. The area of terminal 3 is $a_3$, where there are no available parking slots. Hence, the algorithm provides the next available area, $a_2$ assigning $d_7$ to $s_2$. The same happens to $d_8$, which is assigned to $s_3$.

Essentially, the proposed algorithms are able to associate parking slots to drivers as long as the number of available parking slots in the whole car park is enough to cover the number of drivers.

### 6.6. Complexity

In order to compute the complexity of the indoor parking algorithm, let us compute the cost of the online phases, namely, the fingerprint construction, the area's construction, and the slot association. Let us also recall the set sizes in Table 7, that will be used for the complexity of the procedures.

**Table 7.** Table of set sizes for the complexity.

|  | Set Size |
|---|---|
| Slots | m |
| Access Points | n |
| Terminals | l |
| Drivers | k |

### 6.6.1. Fingerprint Construction

By observing Algorithm 1, one can notice that the first iteration is over the set of slots, with a nested iteration over the set of access points ($m * n$). The second iteration is on the set of terminals, with a nested iteration over the set of access points ($l * n$).

**Proposition 1** (Fingerprint Construction Complexity). *The asymptotic complexity of the fingerprint construction procedure is $O(mn + ln)$.*

### 6.6.2. Area Construction

By observing Algorithm 2, one can notice that the first iteration is on the number of slots ($m$) and the second one is on the number of terminals ($l$). The function *associate_area* has a constant cost. The final iteration is repeated until any slot is associated with an area containing at least one terminal. After a few updates of the threshold, the exit condition is finally true. Hence, this is executed as a constant number $c$ of time over the slots ($c * m$).

**Proposition 2** (Areas Construction Complexity). *The asymptotic complexity of the areas construction procedure is $O(m + l)$.*

### 6.6.3. Slot Association

By observing Algorithm 3, one can notice that the first iteration is on the set of drivers, with a nested *get_available_slot* function depending on the cardinality of the area, which is $m$ in the worst case ($k * m$). The *add* and *remove* functions have constant cost.

One can also observe that the number of drivers is less or equal to the number of slots, as well as the number of terminals and access points, in order that the parking process can be completed.

$$k \leq m \tag{3}$$

$$l \leq m \tag{4}$$

$$n \leq m \tag{5}$$

**Proposition 3** (Slot Association Complexity). *The asymptotic complexity of the slot association procedure is $O(km) = O(m^2)$ by (3).*

**Proposition 4** (Indoor Parking Algorithm Complexity). *The asymptotic complexity of the indoor fingerprint construction procedure is $O(mn + ln) + O(m + l) + O(m^2) = O(m^2 + ln) = O(m^2)$ by (4) and (5).*

## 7. Evaluation

In this section, some experimental results are shown. Precisely, several executions of the algorithm have been tested, with different parameters, shown in Table 8. The number of terminals considered is at most 10, which is already a challenging number, knowing that there is no airport with more than 5 terminals. The number of access points, instead, grows together with the number of vehicles/slots.

**Table 8.** Experimental data.

|  | Terminals | Drivers | Slots | Access Points |
|---|---|---|---|---|
| $E_1$ | 4 | 6 | 9 | 3 |
| $E_2$ | 4 | 20 | 20 | 4 |
| $E_3$ | 5 | 40 | 40 | 10 |
| $E_4$ | 10 | 80 | 80 | 20 |
| $E_5$ | 10 | 150 | 150 | 60 |
| $E_6$ | 10 | 300 | 300 | 100 |
| $E_7$ | 10 | 500 | 500 | 200 |
| $E_8$ | 10 | 1000 | 1000 | 400 |
| $E_9$ | 10 | 3000 | 3000 | 1000 |
| $E_{10}$ | 10 | 10,000 | 10,000 | 3000 |

Performances of the algorithm are shown in Table 9, where execution times of the algorithm are provided for each execution. As we can observe, the vehicle allocation requires a few seconds even in the case of ten thousand drivers/slots. In Figure 3, the time variation can be observed: the number of seconds needed to complete the allocation process is always below 1, except for the benchmark case of 10,000 drivers. Instead, in Figure 4, the algorithm behavior is represented as the variation of time with respect to the parameters previously chosen.

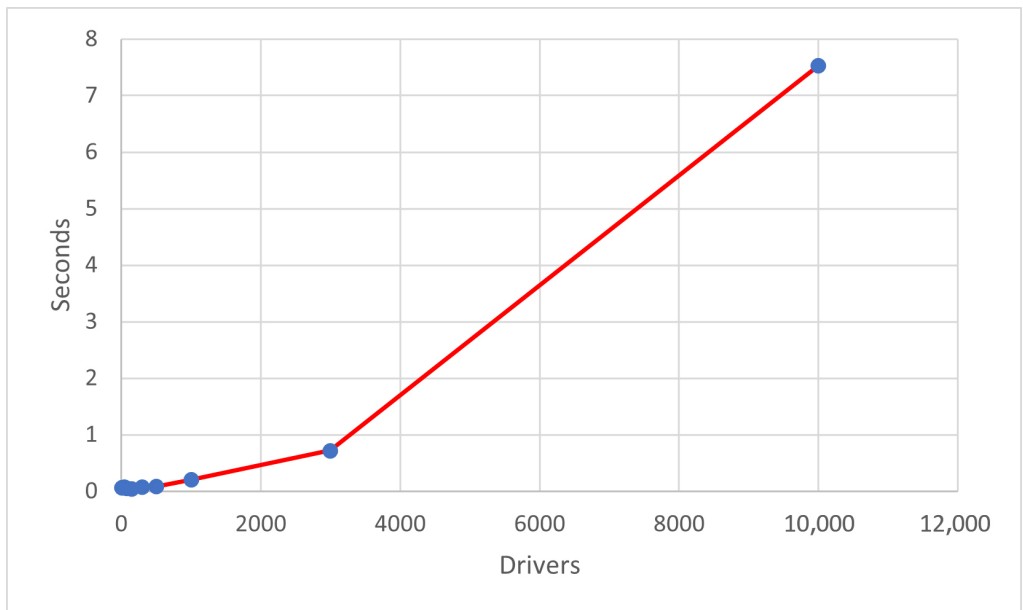

**Figure 3.** Time variation over the number of drivers.

For a matter of completeness, the behavior of the algorithm has been studied for very high numbers of vehicles and slots, solving the problem in 219.843 s for 50,000 drivers/slots, and 938.116 s for 100,000 drivers/slots.

**Table 9.** Performances of the algorithm.

|  | $E_1$ | $E_2$ | $E_3$ | $E_4$ | $E_5$ | $E_6$ | $E_7$ | $E_8$ | $E_9$ | $E_{10}$ |
|---|---|---|---|---|---|---|---|---|---|---|
| seconds | 0.056 | 0.064 | 0.070 | 0.047 | 0.044 | 0.072 | 0.086 | 0.197 | 0.718 | 7.527 |

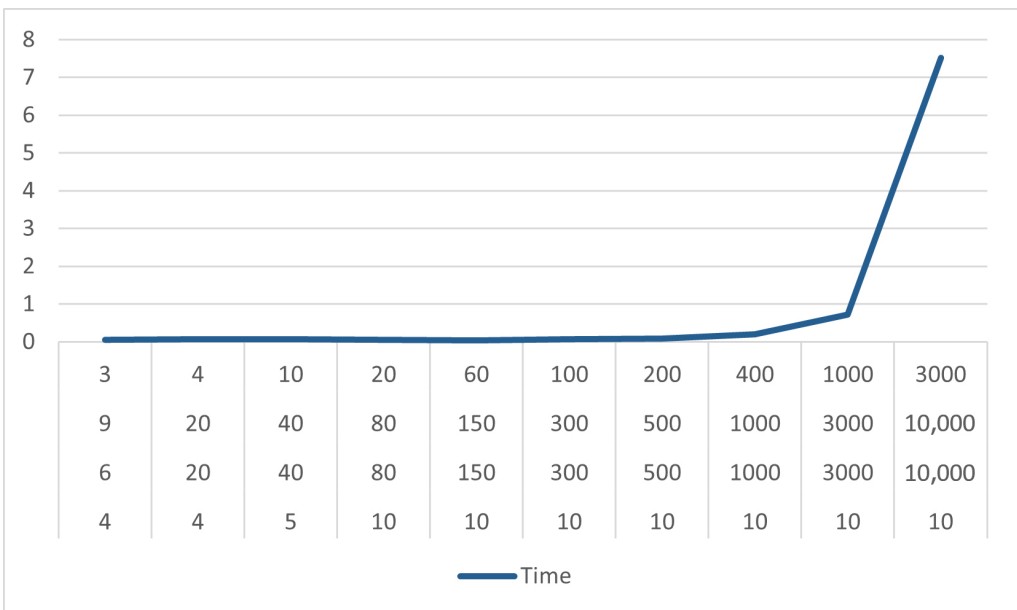

**Figure 4.** Time variation over the number of terminals, drivers, slots and access points.

## 8. Conclusions

The parking problem is still an open issue in the automotive research field, and it is about suggesting suitable parking slots to the drivers that ask for them. This process becomes harder when moving to an indoor environment, where the localization cannot be obtained through a global positioning system.

In this work, the indoor parking problem is faced. First, the environment is opportunely set and, precisely, the problem is defined in a car park of an airport. Then, the indoor parking algorithm is defined, through three phases, sampling, user profiling, fingerprint construction, area construction, and the driver-slot association phase. Each phase is explained by means of a running example and the final solution is a vehicle allocation that satisfies the driver's needs about their final destination. The resulting algorithm works in quadratic time over the number of parking slots. Experimental results prove the efficiency of the algorithm, even with high magnitude numbers of vehicles and slots.

Recently, efficient algorithms based on game theory have been developed for the localization [28–33]. However, they only consider deterministic environments. As a future hint, one could extend the same algorithms to non-deterministic environments by also exploiting techniques of formal methods for strategic reasoning, by using [34–36] as a starting point.

**Funding:** This research received no external funding.

**Institutional Review Board Statement:** Not applicable.

**Informed Consent Statement:** Not applicable.

**Data Availability Statement:** Not applicable.

**Conflicts of Interest:** The author declares no conflict of interest.

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
