# Peer review of "An Indoor Smart Parking Algorithm Based on Fingerprinting"

_futureinternet, doi:10.3390/fi14060185_

Round 1

Reviewer 1 Report

(1) Author should contain traditional algorithms in other papers and be shown its performance exceeding them.

(2) Algorithm should be revised more due to its high complexity, O(n^2), and especially it has a very long time to recommend parking slot.

(3) Moreover, other aspects of experiments should be added to demonstrate the performance of the proposed algorithm (we recommend to add least 3 or 4 graphs including other schemes that were suggested in other papers).

Author Response

(1) In section 2, second paragraph, you can find several solutions for indoor smart parking from the literature. Although, in my opinion, none of them can be directly compared with the solution provided in the paper, or the comparison would be meaningless. Indeed, the authors of those papers propose solutions with a totally different spirit and approach.

(2) Despite the quadratic asymptotic complexity, the algorithm provides a vehicle allocation in less than one second for almost the totality of the experiments. Only the case of 10000 drivers takes 7 seconds, but this is a case that has been reported only for benchmarks and does not reflect a realistic scenario.

(3) Thank you for the interesting suggestion.

Reviewer 2 Report

The authors have partially addressed my previous concern about RSSi values not being reliable.  They have only provided one reference to back up their claim that this is not important.  I would only ask that they provide a few more references to back up their claim before the manuscript can be considered for publication.

Author Response

Thank you for reading the updates. Other references will be provided.

Reviewer 3 Report

Thanks for your revision! You have sufficiently addressed my questions. 

Author Response

Thank you for your attention.

This manuscript is a resubmission of an earlier submission. The following is a list of the peer review reports and author responses from that submission.

Round 1

Reviewer 1 Report

The author addresses the problem of smart parking. In this paper, the algorithm recommends a parking slot based on a driver’s preferences.
However, the scenario should be revised as reflecting a complex airport or terminal by considering the enormous number of parking slots.
Moreover, it did not contain the results of evaluation or simulation, which should prove the performance of the proposed algorithm.
The algorithm should be compared with traditional algorithms in other papers and be shown its performance exceeding them.

Reviewer 2 Report

This manuscript describes an algorithm for allocating parking spots for vehicles in a car park near an airport.  The paper is generally well-written and easy to understand.  However, there is one major issue that has to be addressed before it could be considered for publication.  The proposed algorithm relies heavily on received RSSI values.  The issue is that such values are known to be very unreliable and unpredictable and can be affected for example if the vehicle is blocked from the base station by some obstacle which weakens or stops the transmission to the vehicle.  As a result, the obtained values are unreliable, and so the proposed algorithm may also give unreliable results.  The paper must address this point sufficiently well so that it may be considered for publication.

Reviewer 3 Report

In the last few years, researchers from many research fields are investigating the problem affecting all the drivers in big and populated cities: the parking problem. In the outdoor environment, the problem can be solved by relying on vehicular ad hoc networks, which guarantee communication among vehicles populating the network. When it comes to indoor settings, the problem gets harder, since drivers cannot count on classic GPS localization. In this work, a smart parking solution for a specific indoor setting is provided, exploiting the fingerprint approach for indoor localization. The considered scenario is a multi-level car park inside an airport building. The algorithm provides a vehicle allocation inside the car park in quadratic time over the number of parking slots, by also considering the driver’s preferences on the terminal to be reached.

The literature review is comprehensive. 

In Section 6.4, the area construction uses the maximum signal strength to classify the parking area, which means the parking area is in a circle shape. Slots and terminals in opposite directions but the same distance to an access point in the same area will influence the accuracy of the slot distribution. Can the area construction algorithm be improved to deal with area construction with different shapes?

In Section 6.5, the function ‘get_abailable_slot(area) gives a random available slot in the selected area. There is highly possible that drivers planning to park near t1 and t2 be associated with slots s9 and s1 respectively. How does the proposed algorithm deal with this situation?

Machine learning can be used in fingerprinting to get a more accurate position of the vehicle.

The driver-slot association problem should be solved by a mathematical optimization tool.

Typos:

In Line 242 and 244, there should not be a 3,4,5 over the equal sign.

Reviewer 4 Report

Regarding this paper, the following points must be addressed:

  1. The reason why fingerprinting and rssi are selected instead of other approaches which providing much better localization results is not stated in the paper.
  2. The main challenges/differences between finding parking slot and normal indoor localization are not given in the paper.
  3. The references regarding the state of the art are lacking.
  4. There are no results showing the performance of the proposed algorithm.